# An Unsupervised Image Enhancement Framework for Multiple Fault Detection of Insulators

**DOI:** 10.3390/s25227071

**Published:** 2025-11-19

**Authors:** Jiaxin Guo, Gujing Han, Min He, Yu Li, Liang Qin, Kaipei Liu

**Affiliations:** 1School of Electronics and Electrical Engineering, Wuhan Textile University, Wuhan 430200, China; 2415363074@wtu.edu.cn (J.G.); gjhan@wtu.edu.cn (G.H.); 1995033@wtu.edu.cn (Y.L.); qinliang@whu.edu.cn (L.Q.); kpliu@whu.edu.cn (K.L.); 2Department of Electrical Engineering and Automation, Wuhan University, Wuhan 430072, China

**Keywords:** complex lighting, image enhancement, insulator fault detection, grayscale attention mechanism

## Abstract

To address the problem of low detection accuracy caused by uneven brightness distribution in transmission line inspection images under complex lighting conditions, this paper proposes an unsupervised image enhancement method that integrates grayscale feature guidance and luminance consistency loss constraint. First, a U-shaped generator combining a bottleneck structure with large receptive field depthwise separable convolutions is designed to efficiently extract multi-scale features. Second, a grayscale feature-guided image generation module is incorporated into the generator, using grayscale information to adaptively enhance local low-light regions and effectively suppress overexposed regions. Meanwhile, to accommodate the characteristics of unpaired data training, a luminance consistency loss is introduced. By constraining the global luminance distribution consistency between the generated image and the reference image, the overall brightness balance of the generated image is improved. Finally, a multi-level discriminator structure is constructed to enhance the model’s ability to distinguish global and local luminance in the generated images. Experimental results show that the proposed method significantly improves image quality (PSNR increased from 7.73 to 18.41, SSIM increased from 0.43 to 0.85). Furthermore, the enhanced images lead to improvements in defect detection accuracy.

## 1. Introduction

Power transmission lines are a cornerstone of modern infrastructure, responsible for delivering electrical energy from generation sources to end users. As critical components, insulators electrically isolate conductors from supporting structures, thereby preventing leakage currents and maintaining the safety and stability of the grid. However, faults in insulators can significantly threaten the reliability of power systems. Such defects often arise from environmental contamination, mechanical damage, or flashover events. Therefore, systematic inspection and maintenance of insulators are essential to sustain their performance and ensure reliable operation under diverse environmental conditions.

Deep learning, particularly in the field of computer vision, has significantly advanced various industrial applications, including those in the power sector [1,2,3]. A common method involves using UAVs to capture transmission line inspection images, which are then processed by AI algorithms for fault detection and localization [4]. When datasets are insufficient, many studies enrich their data by transferring target objects into diverse scenes. For example, TAO [5] implemented a cascaded CNN on the CPLID (Chinese Power Line Insulator Dataset) by embedding the same insulator targets into various backgrounds, achieving a precision of 91% and recall of 96%. Similarly, Jiang [6] developed a weakly supervised segmentation-based fusion detection algorithm, using YOLOv5 after segmenting and migrating target objects. Although these data augmentation strategies enhance sample variety, they still struggle to meet the complexity demands of re-al-world scenarios due to limited sample diversity.

When processing large-scale datasets, researchers primarily focus on factors that influence detection accuracy, particularly complex backgrounds and multi-scale defects. Attention mechanisms are commonly employed to distinguish targets from background clutter, emphasizing salient regions while suppressing irrelevant information [7,8]. For example, Zhou [9] integrated a Convolutional Block Attention Module (CBAM [10]) into YOLOv5, using max-pooling and average-pooling at both channel and spatial levels, and achieved a 98% average detection accuracy for glass insulator targets. Similarly, Hao [11] incorporated a CSPResNeSt backbone and a three-dimensional attention module (SiMAM [12]) into YOLOv4, improving insulator detection accuracy by 3.5% in complex back-grounds. However, these methods primarily focus on detecting intact insulators rather than identifying faults. Detecting small-scale fault regions remains challenging due to variations in target size resulting from differences in capture distance. Such diminished targets often degrade model performance. To address this, researchers optimize feature fusion modules for multi-scale integration. Zhao [13], for instance, employed a Feature Pyramid Network (FPN [14]) to enhance Faster-RCNN [15] and improve insulator localization accuracy. Zhang [16], working with YOLOv3, introduced a multi-scale dense connection strategy that preserves critical feature details during convolution, improving small-target detection by retaining finer-scale features within the same stage rather than mixing features across different stages.

Despite the advancements in enhancing detection accuracy and handling complex backgrounds, addressing uneven illumination in challenging remains a major technical bottleneck. Under conditions such as nighttime or adverse weather, conventional visible light imaging often fails to provide sufficient quality for accurate insulator detection. Moreover, environmental factors including pollution, snow, and fog further degrade image visibility, making low-light detection particularly difficult. As noted in [17], research on weak-light enhancement for UAV aerial images of power transmission lines is limited. Although a contrast-adaptive histogram equalization model was proposed, which separates color and brightness through color-space con-version and balances contrast parameters to suppress noise, its robustness in real-world complex conditions remains uncertain.

To handle low-light complexity, deep learning-based methods have emerged. For instance, Liu’s IA-YOLO algorithm [18] and Zhang’s causality-based adaptive detection algorithm [19] preprocess low-light images before performing target detection. These approaches yield high accuracy but suffer from time-consuming preprocessing steps, reducing computational efficiency. Additionally, Zhang’s [20] approach for foggy weather scenarios multiplies the dataset size by a factor of six to simulate various fog concentrations and then applies an improved YOLOv5 model for detection. While this data augmentation technique increases image quantity, it raises concerns about potential overfitting and lacks an explicit, robust preprocessing step tailored for foggy conditions. Thus, there remains a pressing need for methods that effectively enhance image quality under uneven illumination without sacrificing efficiency or generalizability.

To address these challenges, this paper investigates the aforementioned issues in depth. First, we propose an insulator fault detection method for complex environments that integrates low-light enhancement and overexposure correction. The method mitigates problems of uneven illumination by designing a deep-learning–based adversarial network to preprocess raw images, thereby enhancing the visual quality of target regions. Furthermore, the object detection algorithm is optimized to handle complex backgrounds and small-scale targets, ultimately improving the accuracy of insulator fault detection under harsh environmental conditions. The main contributions of this work are summarized as follows:(i)A fault detection model for insulators in transmission lines under low-light and high-exposure environments is developed, covering four categories: normal, self-explosion, damage, and flashover;(ii)A grayscale feature-guided image generation structure is proposed to achieve fine-tuning of the brightness of local areas of the image through grayscale features;(iii)For unpaired data training scenarios, a brightness consistency constraint loss is constructed to improve the balance of the overall brightness distribution of the image by constraining the global brightness of the generated image and the reference image.

## 2. Model Overall Structure Design

The overall structure is shown in Figure 1. In this chapter, a generator model is first designed by combining the U-Net architecture with a bottleneck feature extraction module (FEM) to efficiently capture multi-scale features under complex lighting conditions. The generator employs a combination of 1 × 1 convolution and 5 × 5 depthwise separable convolution, which both expands the receptive field and effectively reduces the model’s computational complexity. To further optimize the generator’s adaptability to complex lighting conditions, an attention mechanism guided by grayscale features is designed, enhancing luminance in target regions while suppressing luminance in background regions, thereby achieving adaptive local brightness adjustment across the image. Additionally, for the generated images, a composite loss function consisting of adversarial loss, self-feature preservation loss, and luminance consistency loss is constructed. The adversarial loss is used to optimize the overall quality of the generated images, while the self-feature preservation loss and luminance consistency loss ensure that the brightness and structural features of the enhanced images are not significantly distorted. This enables the enhancement of visual quality while preserving critical detail features. Finally, multi-patch discrimination is performed on the generated images to ensure consistency in both local details and overall luminance. Based on the above structural design and optimization methods, this chapter proposes an illumination enhancement method that is both efficient and generalizable. It effectively mitigates the negative impact of complex lighting conditions on the quality of transmission line inspection images, significantly enhancing the recognizability of target regions under both overexposed and low-light scenarios. Moreover, it provides stable, high-quality image inputs that facilitate subsequent fault detection tasks. The following sections describe the detailed design and optimization process of the proposed model.

### 2.1. Deep Bottleneck Network Generator and Multi-Patch Discriminator

(1)Generator encoding and decoding structure design

In the processing of transmission line inspection images, complex lighting conditions are primarily characterized by brightness imbalance caused by the coexistence of overexposed and low-illumination regions. This imbalance leads to the loss of local details in images, which in turn severely affects feature extraction and the accurate localization of fault regions. Therefore, the design of the generator structure must effectively balance the extraction of local details and luminance features to achieve brightness balancing and enhancement. To this end, the U-Net architecture, known for its strong feature extraction and reconstruction capabilities, is selected as the basic framework for the generator, combined with a bottleneck feature extraction module to further optimize feature representation. The U-Net framework, with its distinctive encoder–decoder architecture, can simultaneously capture and reconstruct local details, making it particularly effective for image enhancement tasks [21]. During the encoder stage, the model progressively downsamples the input image to obtain feature representations at different scales; in the decoder stage, it gradually upsamples the features to restore detailed information of the image.

To address the problem of high computational complexity in the traditional U-Net structure, this study designs a bottleneck feature extraction module, as shown in Figure 2. By combining 1 × 1 convolutions with 5 × 5 depthwise separable convolutions, the module achieves dimensionality reduction along the channel dimension and improves computational efficiency. Specifically, 1 × 1 convolutions are first used to reduce the channel dimension of the feature map, thereby compressing the feature space and lowering the computational complexity of subsequent convolution operations. Then, 5 × 5 depthwise separable convolutions are employed to expand the receptive field while maintaining low computational cost, effectively capturing a broader range of local detail features. This combination not only reduces model complexity but also enhances the model’s ability to perceive regions with high exposure or low illumination. During the encoding and decoding processes, the model captures multi-scale features through downsampling operations and progressively restores image details through upsampling operations. In the downsampling stage, the model compresses image dimensions to enhance high-level feature representation of brightness distribution; in the upsampling stage, deep features and shallow detail features are effectively fused through skip connections to restore and reconstruct the fine structure of the image. The overall network structure is illustrated in Figure 3.

(2)Discriminator structure design

To further improve the quality of brightness enhancement in the generated images and to avoid detail distortion or local over-enhancement, this study introduces a multi-patch discriminator structure. The discriminator evaluates both the global and local regions of the generated images to ensure consistency in brightness enhancement at both the overall and local levels. The discriminator network adopts VGG as the backbone structure and integrates judgments based on both the global and local views of the image to enhance the overall assessment of brightness quality [22].

Specifically, the global discriminator focuses on evaluating the brightness balance across the entire image, ensuring that the overall brightness distribution and visual effect are consistent, and preventing issues such as overexposure or excessive darkness. The local discriminator randomly crops patches from the image and specifically evaluates the detail enhancement effects in high-exposure or low-light areas, ensuring that local details in the generated images are realistic without obvious distortions or artifacts. The collaborative operation of the global and local discriminators effectively guarantees the quality of brightness enhancement across different scales and regions, significantly improving the model’s adaptability to complex lighting scenarios and enhancing the realism and detail preservation of the output images. As shown in Figure 4, by introducing these two discriminators, the generated images receive more comprehensive supervision regarding their brightness effects, preventing over-enhancement or uneven enhancement issues either globally or locally.

### 2.2. Grayscale Feature-Guided Brightness Uneven Area Enhancement Algorithm

The model structure designed in the previous section primarily focuses on image generation and discrimination but does not specifically model the characteristics of lighting distribution. As a result, it still exhibits insufficient handling of local high-exposure and low-illumination regions, especially in areas with high contrast, where brightness imbalance remains evident. As shown in Figure 5, the HSV3D (Hue-Saturation-Value) visualization indicates that low-light images suffer from insufficient brightness, while high-exposure images exhibit excessively high brightness. When dealing with such complex scenes, the accurate extraction and effective processing of luminance information are critical for determining the quality of image enhancement. Therefore, this section further proposes an uneven brightness optimization algorithm based on grayscale feature guidance. By leveraging the advantages of low-dimensional representation in grayscale features, the algorithm aims to more accurately identify and focus on high-exposure and low-illumination regions, enabling adaptive local brightness adjustment and enhancement. This approach effectively addresses the problem of uneven brightness distribution in images.

Specifically, the proposed algorithm first converts the RGB image into a grayscale image to extract luminance distribution information, thereby avoiding interference from color channels and simplifying the subsequent enhancement process. Next, a grayscale attention-guided mechanism is designed, which effectively improves the model’s focus on regions with uneven brightness through grayscale feature weighting. For low-illumination areas, the model can precisely enhance the details with lower brightness; for high-exposure areas, it can accurately identify and moderately suppress excessive brightness, ensuring that the enhancement effect remains natural and realistic.

(1)Design of the Grayscale Attention Mechanism

First, by applying a weighted combination of the RGB channel pixel values, the original color image is converted into a grayscale image. The weighting process is based on the human eye’s sensitivity to different color channels [23]. The specific calculation formula is given in Equation (1):(1)$$A_{Gray}=0.299\times I_{R}+0.587\times I_{G}+0.144\times I_{B}$$

Here, *I* denotes the feature map of the initial input image, and *I_R_*, *I_G_*, and *I_B_* represent the feature maps of the red, green, and blue channels, respectively. The weight for the red channel is 0.299, reflecting a moderate sensitivity of the human eye to red; the green channel has a weight of 0.587, the highest, indicating that the human eye is most sensitive to green; and the blue channel has a weight of 0.144, the lowest, indicating lower sensitivity to blue.

As shown in Figure 6, the grayscale values of low-illumination regions in the generated grayscale image are relatively low, and such regions are typically key areas that require enhancement. However, if the original grayscale image is directly used for feature guidance, the model may fail to effectively focus on these low-brightness areas, potentially leading to extreme image enhancement outcomes.

After the inverse mapping operation, regions that originally had low brightness correspond to higher grayscale values in the new feature representation, thereby enhancing the model’s focus on low-illumination areas and facilitating targeted enhancement in subsequent steps. To better integrate the grayscale features with the original image features, the fusion is performed using the formula shown in Equation (2):(2)$$X_{out}=\left\{\begin{array}{c} X_{in}\times sigmoid(A_{Gray}^{*})+X_{in}\\ X_{in}\times sigmoid(A_{Gray}^{*})+sigmoid(X_{in}) \end{array}\right.\begin{array}{c} if\\ if \end{array}\begin{array}{c} mean(X_{in})\;<\;50\\ mean(X_{in})\geq 50 \end{array}$$

Here, $A_{Gray}$ (also denoted as $A_{Gray}^{*}$ in Equation (2)) represents the grayscale attention feature map generated from the RGB image through weighted grayscale conversion and inverse mapping. For consistency, all subsequent equations use the notation $A_{Gray}^{*}$ to denote this variable.

During the fusion process, a sigmoid function is applied to the feature map for nonlinear adjustment. This moderately increases the brightness of low-illumination regions, smoothly enhancing the overall image brightness while preserving original detail information, thus avoiding artifacts and detail loss caused by excessive enhancement.

For high-exposure regions, the sigmoid function similarly compresses pixel values into a reasonable range, preventing further amplification of overexposure while retaining the image’s detailed structures.

(2)Multi-scale grayscale image feature fusion

To further enhance the model’s ability to represent luminance information across different scales within an image, this paper proposes a multi-scale grayscale feature fusion strategy to guide the decoder in better restoring brightness and detail information. Specifically, since the grayscale image is generated based on the brightness distribution of the input image, its initial scale matches that of the input. However, the feature maps within the decoder typically exist at multiple spatial scales. Therefore, before fusing the grayscale image with the decoder’s feature maps, appropriate scale transformations of the grayscale image are necessary to ensure size compatibility. To address this issue, this paper designs a multi-angle, multi-scale downsampling strategy.

As shown in Figure 7, global max pooling is first applied to downsample the grayscale image, emphasizing regions with higher brightness to guide the model in effectively suppressing brightness in overexposed scenes. Second, global average pooling is used for downsampling to preserve and reinforce the overall brightness distribution, helping enhance darker regions in low-light scenarios. Finally, a 3 × 3 convolution with a stride of 2 is used to perform local brightness pattern downsampling, capturing finer-grained local luminance variations and ensuring the preservation of local brightness details during downsampling. Subsequently, the three grayscale feature maps obtained from different angles and scales are fused through weighted aggregation and then combined with the decoder’s feature maps, enabling fine-grained, adaptive brightness adjustment across the image. The fusion process of the multi-scale grayscale feature maps is described by Equation (3).(3)$$A_{Gray\text{-}down}^{*}=MaxPool(A_{Gray}^{*})+AvgPool(A_{Gray}^{*})+Conv_{3\times 3}^{2}(A_{Gray}^{*})$$

### 2.3. Loss with Self-Feature Preservation and Brightness Consistency Constraint

In unsupervised image enhancement tasks, due to the lack of paired reference images, the effectiveness of model training heavily depends on the rationality and design of the loss functions. This section constructs unsupervised image loss constraints from three perspectives.

(1)Adversarial Loss for Image Generation and Discrimination

The objective of the generator is to produce realistic images, and its loss function is typically used to measure the similarity between the generated images and real images. The traditional discriminator aims to maximize the probability of real images while minimizing the probability of generated images. However, this approach often causes the model to fall into local optima in order to evade detection by the discriminator.

Therefore, beyond simply considering the probability difference between real and generated images, it is necessary to further optimize the loss function to improve the robustness of the generator. Traditional binary cross-entropy loss can lead to instability during the model training process. To address this issue, this paper introduces the mean squared error (MSE) loss, jointly considering the relative probabilities of real and generated images [24], effectively enhancing training stability and improving the quality of generated images. The specific calculation formulas are given in Equations (4) and (5).(4)$$L_{D}^{Global}=E_{Real}[{\left(D(x)-1\right)}^{2}]+E_{Fake}[{\left(D(G(z))\right)}^{2}]$$(5)$$L_{G}^{Global}=E_{Fake}[{\left(D(G(z))-1\right)}^{2}]+E_{Real}[{\left(D(x)\right)}^{2}]$$

In the above formulas, *D*(*x*) represents the output of the discriminator for real data, and this expectation operation focuses on the loss associated with real data. *G*(*z*) denotes the data generated by the generator from noise, and *D*(*G*(*z*)) represents the output of the discriminator for generated data, with this expectation operation focusing on the loss associated with generated data.

Overall, the discriminator’s loss function considers two aspects of expected loss: one for real data and one for generated data. Similarly, the generator’s loss function also considers two aspects of expected loss, corresponding to the difference between generated and real data. This design enables the discriminator and generator to collaborate during training, gradually improving their respective performances. In addition, the model proposed in this paper introduces a local feature-based loss computation, where five randomly cropped patches are used for processing. The computation method for the local loss is consistent with that of the global loss and is denoted as ${Local}_{D}^{L}$ and ${Local}_{G}^{L}$, respectively. The introduction of this local loss further enhances the model’s ability to handle fine details, improving both the local detail quality and the overall consistency of the generated images.

(2)Image self-feature preservation perceptual loss

Although the adversarial training between the generator and discriminator effectively constrains the generator to learn how to produce more realistic enhanced images, the training in this paper is based on unpaired image enhancement. Due to the semantic and visual differences between the reference images and the generated images, it is difficult to impose fine-grained constraints using pixel-level annotations, which may lead to deviations in the generated results. To address this, this paper further introduces perceptual learning of high-level semantic features between the reference and generated images. By preserving the content consistency of high-level semantic features before and after enhancement, the method constrains the consistency of image content during the generation process. The specific calculation is given in Equation (6).(6)$$L_{SFP}(x)=\frac{1}{W_{i,j}H_{i,j}}\sum_{x=1}^{W_{i,j}}\sum_{x=1}^{H_{i,j}}{\left(x-G(x)\right)}^{2}$$

Here, *x* denotes the feature map from the encoder stage, and *G*(*x*) represents the feature map from the decoder stage. *W* and H denote the width and height of the feature maps, respectively, while *i* and *j* refer to the *i*-th pooling layer and the *j*-th convolutional layer in the feature extraction process.

Unlike traditional pixel-level losses, this perceptual loss does not directly compare pixel-wise differences between the generated and target images. Instead, it performs the comparison in a high-level feature space. These features are extracted using a pretrained VGG network, which can capture high-level semantic information of images, such as shapes, textures, and edges. The intermediate convolutional layers are responsible for extracting object contours, structural information, and global features of the image. The features extracted at these layers simultaneously retain local structures and global semantics, enabling the perceptual loss to effectively preserve the geometric structure and content consistency of the images without relying on strict pixel-level comparison.

(3)Image brightness consistency loss

In image enhancement tasks, luminance balance is crucial to the quality of the generated images. In the previously designed model structure, the grayscale attention mechanism was used to guide the model in learning and enhancing uneven luminance. However, relying solely on feature extraction layers for guidance is insufficient to guarantee the global brightness consistency of the generated images, as feature extraction layers primarily focus on high-level semantic information while low-level semantic features also need to be addressed. To solve this problem, this paper further designs a luminance guidance mechanism at the loss function level. The final generated image is used as a source of low-level semantic features to guide global brightness balancing, aiming to ensure that the overall luminance of the generated image matches that of the unpaired target image.

First, global luminance comparison is performed, focusing on the overall luminance trend of the image rather than precise matching of individual pixel brightness. This process ensures that the global luminance of the generated image remains consistent with that of the target image. The specific calculation formula is given in Equation (7).(7)$$L_{Global\_brightness}=|A_{Gray}(G(x_{A}))-A_{Gray}(x_{B})|$$

Here, $A_{Gray}(G(x_{A}))$ and $A_{Gray}(x_{B})$ represent the mean grayscale values of the generated image and the target image, respectively.

Finally, an adaptive luminance constraint is introduced. Unlike methods that forcibly align every generated image with the reference image, the adaptive luminance constraint sets an adaptive target for each image’s luminance distribution. By calculating the dynamic range of the image’s luminance (i.e., the minimum and maximum luminance values), the generated image’s luminance range is matched to that of the reference image. The calculation formula is provided in Equation (8)(8)$$L_{dynamic\_range}=|(\max(A_{Gray}(x_{A}))-\min(A_{Gray}(x_{A})))-(\max(A_{Gray}(x_{B}))-\min(A_{Gray}(x_{B})))|$$

This approach does not require the luminance of every pixel in the generated image to match the reference image exactly. Instead, it focuses on aligning the overall dynamic range of brightness. By constraining the dynamic range of the generated image’s luminance to match that of the reference image, it effectively prevents the generated images from being excessively bright or dark, ensuring the naturalness of the luminance distribution. The corresponding calculation formula is shown in Equation (9)(9)$$L_{Total}=L_{G}^{Global}+L_{G}^{Local}+L_{SFP}+L_{Globale\_brightness}+L_{dynamic\_range}$$

By constructing loss functions based on the structural features and luminance information of the generated images, this approach ensures that the enhanced images preserve their original details and luminance characteristics while improving brightness.

## 3. Experiments and Analysis

### 3.1. Experimental Environment and Data Description

To facilitate the recognition of various fault types in transmission line insulators, this study constructs a dedicated dataset of 4615 images. Each image may contain one or more target instances, including multiple instances of the same fault type. The dataset comprises four categories—normal insulator, self-exploded suspension insulator, broken insulator, and flashover insulator—labeled as Insulator, Defect, Broke, and Flashover, respectively. Figure 8 presents the number of samples in each category. Given the two sets of comparison experiments, the entire dataset is first evenly split. One subset is used for image enhancement training, and the trained model is then evaluated on the other subset. The subsequently enhanced images are finally employed for object detection and localization experiments.

The entire dataset is first used for image enhancement training. The resulting enhanced dataset is then divided into training, validation, and test sets with an 8:1:1 ratio. All experiments are conducted on Ubuntu 20.04 using Python 3.8.0 and CUDA 11.2, in a PyTorch 1.8 environment accelerated by two NVIDIA GeForce RTX 3090-24G GPUs.(The equipment was purchased from Xiandao Feng Information Technology Co., Ltd.) During the enhancement stage, the model is trained for 200 epochs with a batch size of 8, using 512 × 512 resolution images. Optimization adopts a stochastic gradient descent (SGD) strategy with an initial learning rate of 0.01, a minimum of 0.0001, momentum of 0.937, and weight decay of 0.0005. Early stopping is applied if accuracy does not improve for 20 epochs.

### 3.2. Experimental Index

The performance of the proposed model is evaluated using quantitative metrics tailored to both image enhancement and object detection tasks. For image enhancement, peak signal-to-noise ratio (PSNR) and structural similarity index (SSIM) are employed to measure brightness balance and structural fidelity. For object detection, average precision at IoU threshold 0.5 (AP50) and mean average precision (mAP) are used to evaluate detection accuracy across categories. The corresponding calculation formulas are provided below.(10)$$MSE=\frac{1}{I*J}\sum_{i=1}^{I}\sum_{j=1}^{J}{\left[f(i,j)-F(i,j)\right]}^{2}$$(11)$$PSNR=10*\log_{10}(\frac{{\left(2^{n}-1\right)}^{2}}{MSE})$$

Among them, *I*∗*J* represents the length and width of the image, n represents the side length of the image, and *f (i, j)* and *F (i, j)* represent the original image and the enhanced image, respectively.(12)$$\begin{array}{cc} SSIM(X,Y) & =L(X,Y)*C(X,Y)*S(X,Y)\\ & =\frac{\left(2u_{X}u_{Y}+C_{1})(2\sigma_{XY}+C_{2}\right)}{\left(u_{X}^{2}+u_{Y}^{2}+C_{1})(\sigma_{X}^{2}+\sigma_{Y}^{2}+C_{2}\right)} \end{array}$$

Among them, *L(X,Y)*, *C(X,Y)*, and *S(X,Y)* are brightness, contrast, and structural difference functions, respectively; X and Y are the original image and enhanced image, respectively, $u_{X}$ and $u_{Y}$, $\sigma_{X}^{2}$ and $\sigma_{Y}^{2}$, and $\sigma_{Y}^{X}$ are the average value, variance, and covariance of *X* and *Y*, respectively; and C1 and C2 are dynamic constants.

### 3.3. Transmission Line Power Component Defect Enhancement Data Description

To fully enhance the model’s generalization ability and robustness under complex lighting conditions, this paper constructs a synthetic dataset specifically designed to simulate extreme lighting conditions encountered during transmission line inspection. During the image enhancement training process, the entire original dataset is used for simulated extreme lighting condition enhancement, with the synthesized dataset being split in a 1:1 ratio for model training and generalization performance testing.

As shown in Figure 9, Specifically, to simulate extreme lighting scenarios that may be encountered in inspection tasks, this dataset randomly sets all image samples to either overexposure or low-illumination scenes, without retaining any original images with normal lighting conditions. Overexposed samples are synthesized by increasing the overall image brightness, effectively simulating overexposure phenomena caused by direct sunlight or component reflections during transmission line inspections. Low-illumination samples are synthesized by decreasing the overall image brightness to simulate situations such as cloudy days, evening periods, or environments with occlusions, where insufficient brightness may occur. By randomly generating overexposed and low-illumination samples, the model is ensured to undergo comprehensive training under a variety of complex lighting conditions, enabling it to learn a more comprehensive and rich set of visual features.

### 3.4. Image Enhancement Experimental Analysis

#### 3.4.1. Image Enhancement Ablation Experiment Analysis

We compared the performances of several object detection models under complex lighting conditions, aiming to validate the effectiveness of our designed data augmentation method in improving model performance. We used a dataset processed for lighting conditions, as well as complex lighting data generated by our designed model enhancement.

First, we conduct ablation experiments on image enhancement to evaluate the specific contribution of each component to the performance. The experimental results are summarized in Table 1.

The three improved modules designed in this paper—backbone network (Backbone), gray attention mechanism (Gray-Attention), and loss function constraints (Loss)—all have a significant impact on the performance of the image enhancement task. The following is a detailed analysis:

Single Module Enhancement Experiments. Backbone Only: PSNR improves from 7.73 to 12.57, and SSIM from 0.43 to 0.78. This substantial gain indicates that the backbone network greatly enhances feature extraction, resulting in improved structural fidelity and image quality. Gray-Attention Only: PSNR rises to 14.44, and SSIM to 0.81. This demonstrates that the gray attention mechanism effectively addresses uneven illumination issues, leading to a marked improvement in enhancement quality. Loss Only: PSNR improves to 11.65, and SSIM to 0.74. Although the improvement is less pronounced, it positively influences training stability and performance optimization.

Ablation Experiments. Backbone + Loss: PSNR reaches 16.51, and SSIM rises to 0.84, indicating that the combined effect improves detail restoration and structural integrity. Backbone + Gray-Attention: PSNR attains 17.37, and SSIM reaches 0.83, highlighting that the synergy between the backbone and attention mechanism effectively mitigates lighting discrepancies, enhancing overall image quality. Gray-Attention + Loss: PSNR reaches 16.54, and SSIM 0.84, underscoring that this combination further refines processing under uneven lighting compared to either module alone. All Three Combined (Backbone + Gray-Attention + Loss): PSNR achieves 18.41, and SSIM 0.85, revealing a strong synergistic effect. Integrating all three modules yields the highest enhancement performance, improving detail recovery and structural quality.

Overall Summary. Each module addresses a distinct enhancement aspect. The backbone network improves feature extraction, the gray attention mechanism corrects uneven illumination, and the loss function enhances training stability and generalization. When integrated, these modules achieve a synergistic effect that maximizes image enhancement accuracy and quality.

#### 3.4.2. Experiment on Synthetic Image Enhancement with Uneven Illumination

The proposed target enhancement module, which preprocesses the original images, is designed to improve the subsequent transmission line fault detection module’s accuracy. In its initial stage, this module moderately enhances image quality and identifies the insulator region. To evaluate its effectiveness, five advanced image enhancement approaches—Zero-DCE++ [25], CutGAN [26], StyleGAN [27], RetinexNet, EnlightenGAN,—are compared against the proposed target enhancement module. All experiments are conducted using the self-constructed dataset, focusing on both low-illumination and high-exposure images. Comparative results are presented in Table 2 and Table 3. In these tables, the first and last rows show the input images and their corresponding ground-truth label images, while the intermediate rows display results from the various enhancement models.

(1)Enhanced performance analysis for low-light scenes

In low-illumination scenarios, all compared models increase image brightness to varying degrees. However, the unsupervised Zero-DCE++ approach, lacking a brightness reference, suffers from severe color distortion and inconsistency despite effectively brightening the image. RetinexNet also introduces color distortion and heightened noise, resulting in considerable detail loss. EnlightenGAN and CutGAN produce comparatively limited brightness enhancements, leaving some regions insufficiently illuminated. While StyleGAN performs better overall, it still produces uneven brightness, with transmission lines appearing overly bright and backgrounds remaining dark. In contrast, the proposed algorithm substantially improves brightness, contrast, and color saturation, yielding clearer images that facilitate subsequent fault detection. Visually, these results validate the proposed model’s effectiveness in enhancing low-illumination images. The specific renderings are shown in Table 2 below.

**Table 2 sensors-25-07071-t002:** Low-light visualization results analysis.

Model	
ZeroDCE++	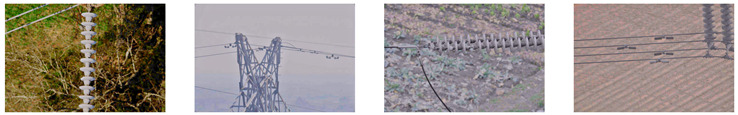
RetinexNet	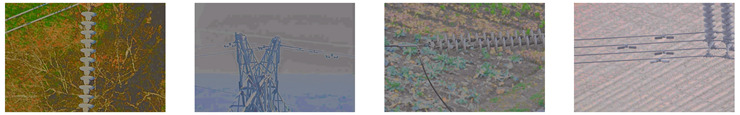
EnlightenGAN	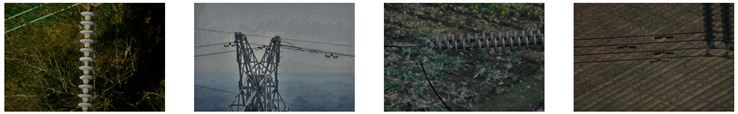
CutGAN	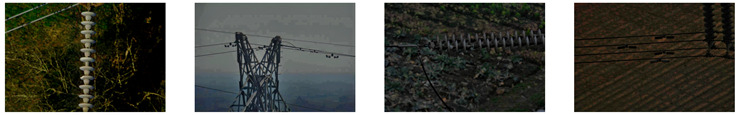
StyleGAN	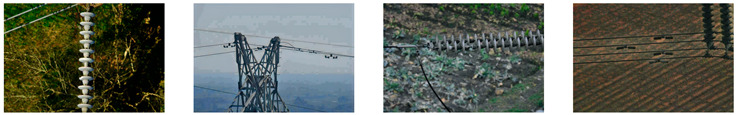
Ours	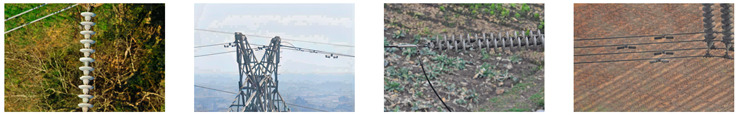
Lables	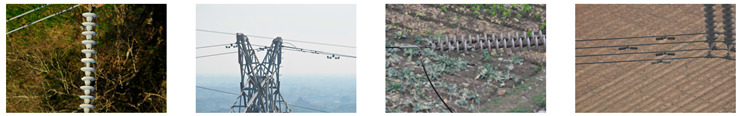

(2)Enhanced performance analysis for high exposure scenes

In high-exposure scenarios, each compared model reduces image brightness, yet their quality improvements differ. Zero-DCE++, as in the low-illumination case, effectively adjusts brightness but introduces severe color distortion and inconsistency. RetinexNet over-darkens the image, leading to over-processing. EnlightenGAN and CutGAN, originally designed for low-illumination enhancement, are less effective for high-exposure images, producing results that gravitate toward a fixed brightness value. While StyleGAN outperforms these methods overall, it still suffers from uneven brightness distribution, with excessively bright transmission lines contrasting with dim backgrounds, as seen in the third image. By contrast, the proposed algorithm mitigates high-exposure effects more effectively, offering balanced brightness and improved clarity. From a visual standpoint, these results confirm the proposed model’s efficacy in enhancing high-exposure images.

**Table 3 sensors-25-07071-t003:** High-exposure visual results analysis.

Model	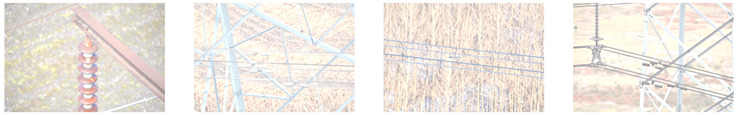
ZeroDCE++	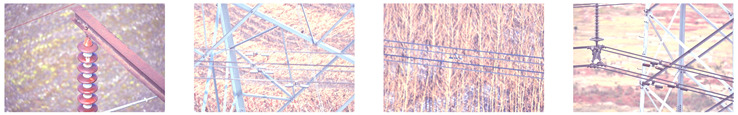
RetinexNet	
EnlightenGAN	
CutGAN	
StyleGAN	
Ours	
Lables	

#### 3.4.3. Experimental Test of Image Enhancement on Actual Power Transmission Lines

In the previous experiments, we used synthetic images with uneven lighting to evaluate the model’s performance. To further validate the model’s performance in real-world scenarios, we introduced actual uneven lighting data from real scenes for testing.

(1)Comparison of actual image detection results

We conducted a comprehensive analysis of various YOLO versions on complex lighting datasets, focusing on three key dimensions: baseline accuracy without the Image Enhancement Module (IEM), accuracy improvements after introducing IEM, and detailed improvements across different detection tasks. The analysis results are summarized in Table 4.

Baseline Accuracy Without IEM: Multiple YOLO variants (v5, v7, v8, v9, and v10) were evaluated on tasks such as insulator, defect, flashover, and damage detection. Earlier versions (e.g., YOLOv5) performed relatively poorly, with YOLOv5s achieving an AP of 94.3% for insulator detection but only 64.9% for flashover detection. As model versions advanced, performance improved, particularly for complex detection tasks. YOLOv8 exhibited the best performance, indicating its suitability for complex lighting conditions in our dataset.

Accuracy Analysis After Introducing IEM: After introducing the Image Enhancement Module (IEM), all YOLO versions showed general improvements in detection performance. Notably, IEM significantly boosted detection accuracy in tasks like insulator and defect detection. Additionally, YOLOv8m+IEM achieved an mAP of 89.8%, surpassing the non-enhanced model’s mAP of 87.8%.

Detailed Accuracy Improvement Analysis: The enhancements varied by detection task. For insulator detection, all YOLO models benefited from IEM. YOLOv5s improved by 0.8 points in mAP, while YOLOv7Tiny saw the largest gain of 4.2 points. The YOLOv8 series improved by 1.7 and 2.0 points, the YOLOv9 series by 0.2 and 1.0 points, and the YOLOv10 series by 1.6 and 2.4 points. These consistent improvements confirm the efficacy of the proposed image enhancement approach.

Although the overall mAP shows an improvement after introducing the illumination enhancement module (IEM), a slight decline in the AP of certain categories (e.g., the flashover class in YOLOv5) can still be observed. This phenomenon may result from the illumination enhancement altering the texture or brightness distribution of specific samples. In particular, for categories like flashover, whose visual features are mainly represented by high-brightness or overexposed regions, excessive enhancement may reduce the feature contrast between target and background, leading to a minor degradation in detection accuracy. Moreover, the IEM focuses on global illumination correction rather than fine-grained texture recovery, which can slightly affect models that rely heavily on local feature cues.

(2)Visual comparison of actual image detection results

Table 5 presents enhanced and detected results using the proposed model on real-world images. In the first row, the original image appears overexposed, while the enhanced version more effectively corrects brightness and yields higher detection confidence. In the second through fourth rows, the images suffer from underexposure, causing some insulator damage to go undetected. After enhancement, brightness is improved, reducing missed detections. In the fifth row, flashover issues on insulators are present against a darker background; the proposed model’s enhancement clarifies the image, increasing flashover detection accuracy. In the sixth row, an overly bright background interferes with flashover detection, resulting in missed cases using the original model. By contrast, the proposed model appropriately reduces brightness, enhancing the flashover detection rate. Overall, these real-world results further confirm the effectiveness of the proposed image enhancement model.

## 4. Discussion

Although the proposed unsupervised image enhancement framework has demonstrated considerable effectiveness for multiple fault detection of insulators under complex lighting conditions, this study has certain limitations that point toward valuable future research directions.

Firstly, in a minority of extreme over-exposed or low-light scenarios, the enhanced images may occasionally exhibit slight color casts. This indicates that achieving better preservation of natural color fidelity while balancing luminance remains an area for further investigation.

Secondly, the dataset constructed for this study suffers from class imbalance, with “normal” insulator samples significantly outnumbering specific fault categories such as “flashover.” While the image enhancement preprocessing improved the overall performance of the detection model, this skewed data distribution might still limit the model’s ability to capture subtle features of the minority fault classes.

Finally, the current work focuses specifically on insulators. The core idea underlying our approach—improving the performance of subsequent detection tasks via unsupervised image enhancement—holds the potential to be transferred to the defect detection of other power equipment (e.g., surge arresters, instrument transformers). Verifying and extending the generalizability of this framework constitutes an important direction for future work.

## 5. Conclusions

To achieve insulator fault detection in complex environments with uneven lighting, especially under high-exposure and low-light conditions, this paper proposes several innovative designs for efficient lighting enhancement strategies. The following conclusions can be drawn:The proposed enhancement strategy can adaptively enhance images with imbalanced lighting under the condition of unpaired data. This method effectively improves the lighting balance of the generated images, preserving local details and global consistency. As a result, the enhanced images are more suitable for subsequent detection and recognition tasks. The PSNR of the enhanced images increases from 7.73 to 18.41, and the SSIM improves from 0.42 to 0.85.Furthermore, validation on a real-world dataset shows that the target detection algorithm, after image enhancement, significantly outperforms direct detection on the raw images. The average precision of the insulator defect detection dataset improves by 0.2% to 4.2% across different models.Finally, the insulator fault dataset constructed in this paper faces a class imbalance issue, which results in high detection accuracy for insulator targets but is also affected by complex backgrounds. Therefore, further research will continue to address the challenges posed by complex backgrounds and targets in complex scenes.

## Figures and Tables

**Figure 1 sensors-25-07071-f001:**
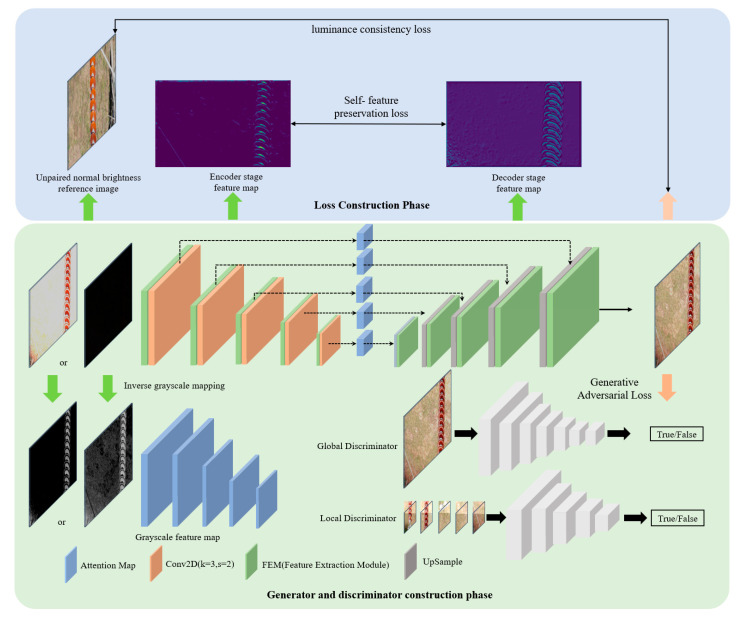
Overall algorithm structure diagram.

**Figure 2 sensors-25-07071-f002:**
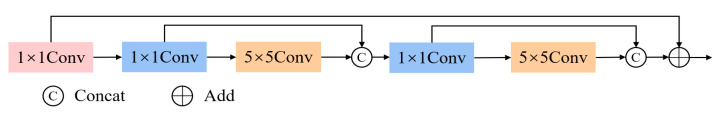
Backbone feature extraction module.

**Figure 3 sensors-25-07071-f003:**
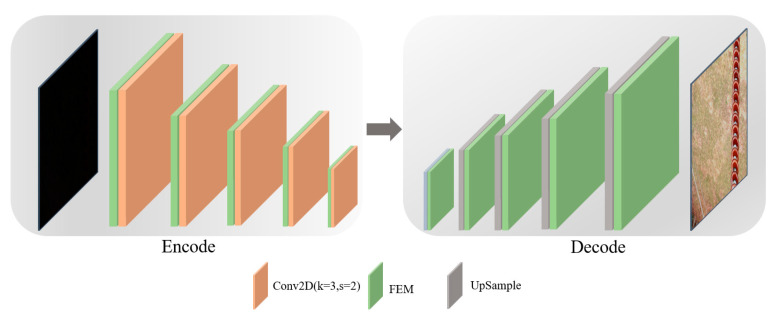
Backbone feature extraction structure.

**Figure 4 sensors-25-07071-f004:**
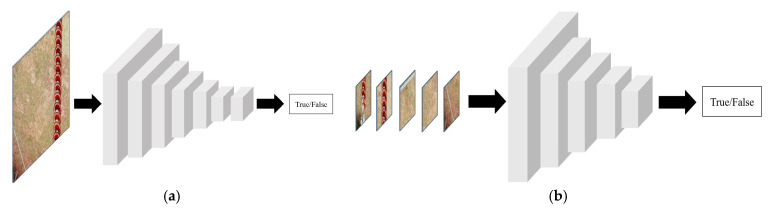
Discriminator network structure. (**a**) Global discrimination. (**b**) Local discrimination.

**Figure 5 sensors-25-07071-f005:**
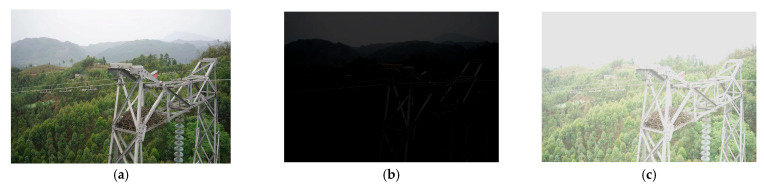
Comparison of normal lighting images and complex lighting images. (**a**) Normal brightness. (**b**) Low illumination. (**c**) High exposure. (**d**) HSV value of normal brightness. (**e**) HSV value at low illumination. (**f**) HSV value of high exposure.

**Figure 6 sensors-25-07071-f006:**
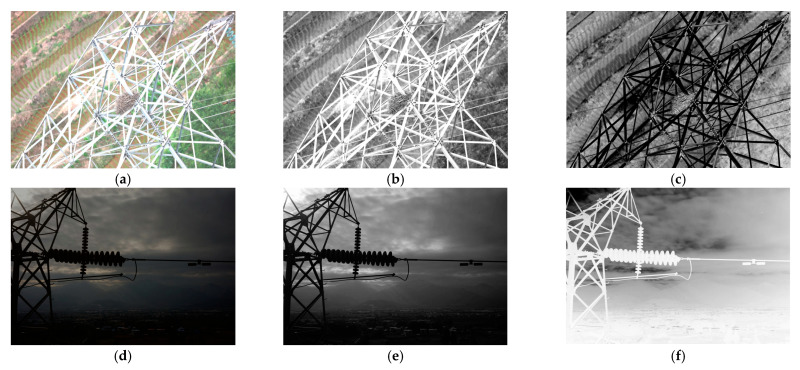
The grayscale and inverse grayscale mapping of an unbalanced brightness image. (**a**) High exposure image. (**b**) Grayscale image mapping with high exposure. (**c**) High exposure reverse grayscale image mapping. (**d**) Low exposure image. (**e**) Low-exposure grayscale mapping. (**f**) Low exposure reverse grayscale mapping.

**Figure 7 sensors-25-07071-f007:**
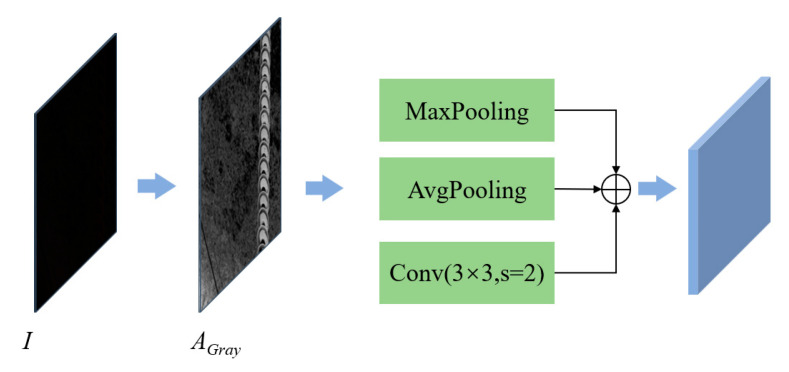
Grayscale attention feature fusion structure.

**Figure 8 sensors-25-07071-f008:**
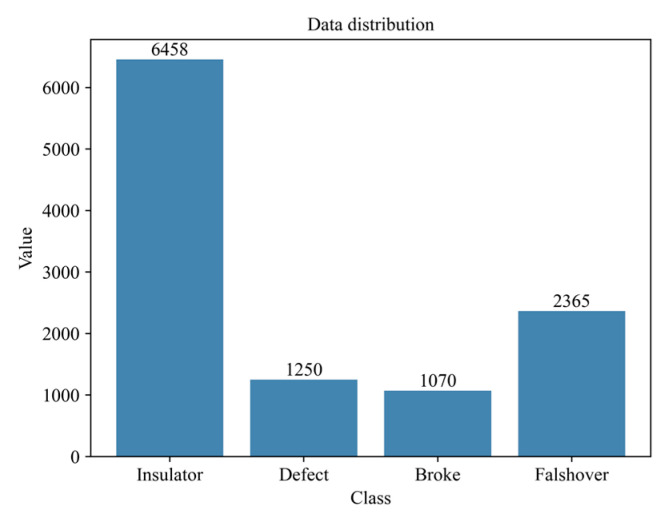
Data distribution chart.

**Figure 9 sensors-25-07071-f009:**
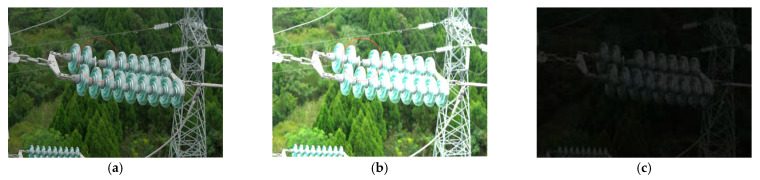
Comparison of images with different brightness levels. (**a**) Original image. (**b**) Overexposed image. (**c**) Low-light image.

**Table 1 sensors-25-07071-t001:** Ablation Experiment Analysis on Synthetic Data.

Backbone	Gray-Attention	Loss	PSNR (dB)	SSIM
-	-	-	7.73	0.43
√	-	-	12.57	0.78
-	√	-	14.44	0.81
-	-	√	11.65	0.74
√	-	√	16.51	0.84
√	√	-	17.37	0.83
-	√	√	16.54	0.84
√	√	√	18.41	0.85

**Table 4 sensors-25-07071-t004:** Accuracy comparison of insulator and defect datasets before and after enhancement.

Model	AP/%				mAP/%
	Insulator	Defect	Flashover	Broke	
YOLOv5s	94.3	89.8	64.9	73.6	80.6
YOLOv5s + IEM	94.6	91.3	63.8	76.0	81.4**^+0.8^**
YOLOv5m	94.9	92.3	73.1	86.4	86.7
YOLOv5m + IEM	96.0	90.8	72.7	87.7	86.8**^+0.1^**
YOLOv7Tiny	90.5	80.8	54.0	63.6	72.2
YOLOv7Tiny + IEM	95.7	91.8	68.5	73.5	79.4**^+4.2^**
YOLOv8s	95.0	90.9	65.3	78.3	82.3
YOLOv8s + IEM	94.8	91.9	70.9	78.3	84.0**^+1.7^**
YOLOv8m	95.3	92.9	74.4	88.7	87.8
YOLOv8m + IEM	95.7	93.2	77.9	92.4	89.8**^+2.0^**
YOLOv9s	94.3	88.1	61.1	75.3	79.7
YOLOv9s + IEM	93.8	86.3	60.1	79.3	79.9**^+0.2^**
YOLOv9m	95.1	92.7	72.5	88.7	87.3
YOLOv9m + IEM	95.7	93.4	74.7	89.3	88.3**^+1.0^**
YOLOv10s	92.8	88.0	61.3	67.7	77.4
YOLOv10s + IEM	93.8	87.9	62.8	71.4	79.0**^+1.6^**
YOLOv10m	93.5	87.2	64.7	69.3	78.7
YOLOv10m + IEM	93.6	89.8	67.8	73.1	81.1**^+2.4^**

**Table 5 sensors-25-07071-t005:** Visual comparison of actual image detection results.

Original Image	Original Image Detection Results	Enhanced Images	Enhanced Image Detection Results
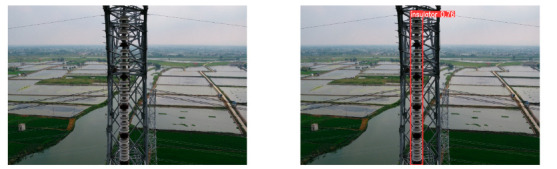		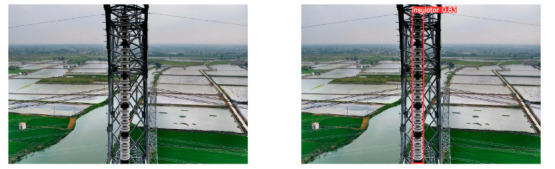	
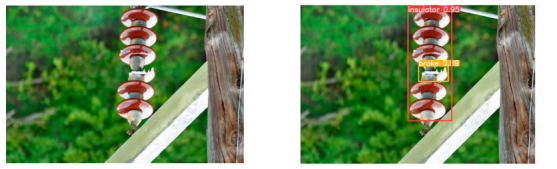		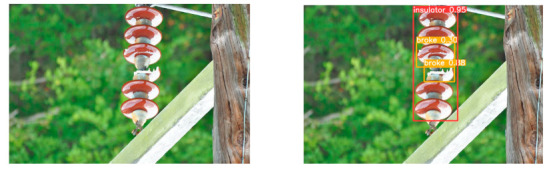	
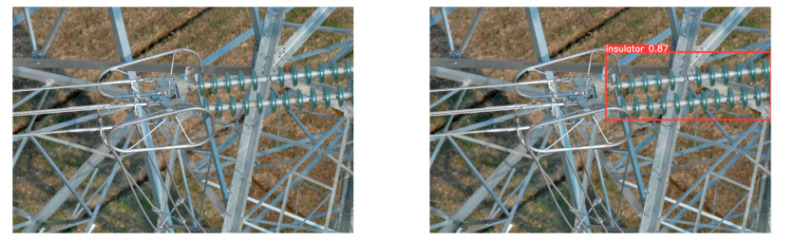		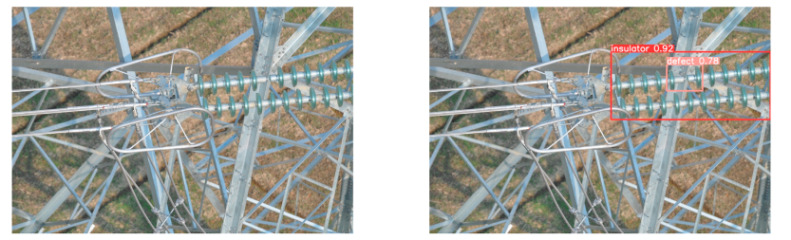	
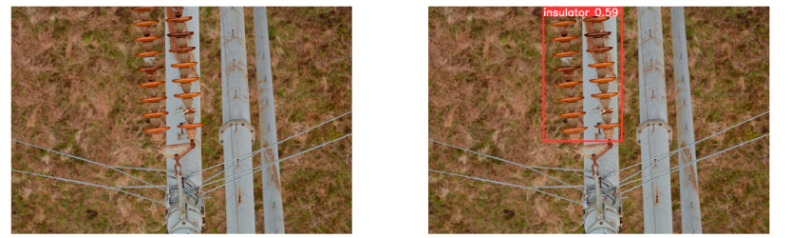		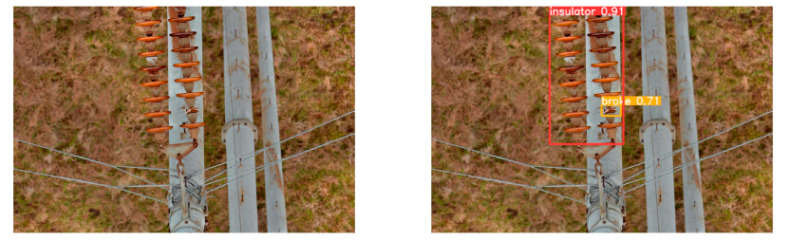	
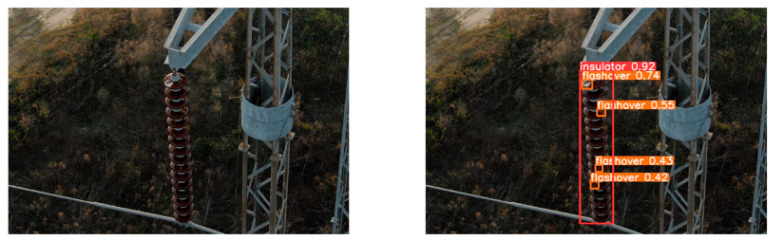		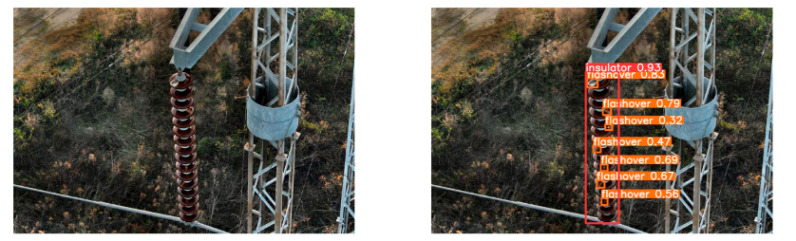	
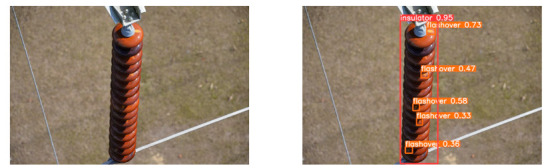		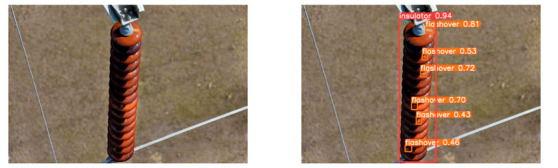	

## Data Availability

Embargo on data due to commercial restrictions.

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
