# Peer review of "An Unsupervised Image Enhancement Framework for Multiple Fault Detection of Insulators"

_sensors, 2025, doi:10.3390/s25227071_

Round 1
Reviewer 1 Report
Comments and Suggestions for Authors
This paper proposes an unsupervised enhancement framework for transmission-line insulator images to address the decline in detection accuracy under low-light and overexposure conditions. The overall structure is clear. The main innovations lie in introducing a grayscale-guided brightness enhancement mechanism and a brightness consistency loss, and employing a multi-scale discriminator to improve detail reconstruction in generated images. The method significantly improves detector accuracy under unpaired data settings and has practical engineering value. Overall, the content is complete, the experiments are sufficient, and the conclusions are reasonable; however, the literature review, experimental details, equation explanations, and discussion sections still require strengthening.
1. The introduction clearly presents the background and challenges but lacks citations to classic methods, such as CBAM, SiMAM (typical attention mechanisms), and Faster R-CNN (a classic detection framework). It is recommended to add these references in the introduction.
2. In the experiments, please provide the missing hyperparameter settings—especially for the image enhancement component. In addition, the parameter descriptions in Section 3.2 overlap with the experimental settings in Section 3.1; please revise to remove redundancy.
3. Several equations lack necessary explanations. For example, in Equation (2), the source of A*gray is unclear. Please provide detailed definitions for all variables appearing in the formulas.
4. Add a short Discussion section to articulate current limitations and outline directions for future research.
Author Response
This paper proposes an unsupervised enhancement framework for transmission-line insulator images to address the decline in detection accuracy under low-light and overexposure conditions. The overall structure is clear. The main innovations lie in introducing a grayscale-guided brightness enhancement mechanism and a brightness consistency loss, and employing a multi-scale discriminator to improve detail reconstruction in generated images. The method significantly improves detector accuracy under unpaired data settings and has practical engineering value. Overall, the content is complete, the experiments are sufficient, and the conclusions are reasonable; however, the literature review, experimental details, equation explanations, and discussion sections still require strengthening.
Comments1. The introduction clearly presents the background and challenges but lacks citations to classic methods, such as CBAM, SiMAM (typical attention mechanisms), and Faster R-CNN (a classic detection framework). It is recommended to add these references in the introduction.
Response1: We appreciate the reviewer’s valuable suggestion. In the revised manuscript, we have added citations and brief discussions of the representative attention mechanisms (CBAM, SiMAM,FPN) and the classical detection framework (Faster R-CNN) in the Introduction (Section 1, lines 52–63).
Add references as shown below, marked in blue font in the original text.
- S Woo, J Park, JY Lee. “Cbam: Convolutional block attention module”Proceedings of the European conference on computer vision (ECCV). 2018: 3-19.
- L Yang, RY Zhang, L Li. “Simam: A simple, parameter-free attention module for convolutional neural networks”International conference on machine learning. PMLR, 2021: 11863-11874.
- T Y Lin, P Dollár, R Girshick, K He, B Hariharan, S Belongie, “Feature pyramid networks for object detection”. In Proc. IEEE Conf. Comput. Vis. Pattern Recognit. (CVPR), Honolulu, HI, USA, 21–26 July 2017; pp. 2117–2125.
- R Girshick. “Fast R-CNN”. In Proc. IEEE Int. Conf. Comput. Vis. (ICCV), Santiago, Chile, 7–13 December 2015; pp. 1440–1448.
Comments 2. In the experiments, please provide the missing hyperparameter settings—especially for the image enhancement component. In addition, the parameter descriptions in Section 3.2 overlap with the experimental settings in Section 3.1; please revise to remove redundancy.
Response2: Thank you for this insightful comment. We have added detailed hyperparameter settings for the image enhancement module in Section 3.1, including the learning rate (0.01 with dynamic adjustment to 0.0001), batch size (8), and number of epochs (200). To eliminate redundancy, we have simplified Section 3.2 to retain only the metric definitions (PSNR, SSIM) and removed the overlapping training parameter descriptions. The revisions are highlighted in Section 3.1 (lines 376–384) and Section 3.2 (lines 388–393). The revised original text is as follows.
The entire dataset is first used for image enhancement training. The resulting en-hanced dataset is then divided into training, validation, and test sets with an 8:1:1 ratio. All experiments are conducted on Ubuntu 20.04 using Python 3.8.0 and CUDA 11.2, in a PyTorch 1.8 environment accelerated by two NVIDIA GeForce RTX 3090-24G GPUs. During the enhancement stage, the model is trained for 200 epochs with a batch size of 8, using 512×512 resolution images. Optimization adopts a stochastic gradient descent (SGD) strategy with an initial learning rate of 0.01, a minimum of 0.0001, momentum of 0.937, and weight decay of 0.0005. Early stopping is applied if accuracy does not im-prove for 20 epochs.
The performance of the proposed model is evaluated using quantitative metrics tailored to both image enhancement and object detection tasks. For image enhancement, peak signal-to-noise ratio (PSNR) and structural similarity index (SSIM) are employed to measure brightness balance and structural fidelity. For object detection, average pre-cision at IoU threshold 0.5 (AP50) and mean average precision (mAP) are used to eval-uate detection accuracy across categories. The corresponding calculation formulas are provided below.
Comments 3. Several equations lack necessary explanations. For example, in Equation (2), the source of A*gray is unclear. Please provide detailed definitions for all variables appearing in the formulas.
Response3: We sincerely appreciate the reviewer’s careful observation. In the revised manuscript (Section 2.2), we have added explicit definitions and clarifications for all variables. Specifically, for Equation (2), we now clarify that A_gray and A*gray refer to the same variable. This variable denotes the grayscale attention feature map generated from the RGB image through weighted grayscale conversion and inverse mapping, which guides the model to enhance brightness in low-illumination regions. The revised original text is as follows.
Here, (also denoted as in Equation (2)) represents the grayscale attention feature map generated from the RGB image through weighted grayscale conversion and inverse mapping. For consistency, all subsequent equations use the notation to denote this variable.
Comments 4. Add a short Discussion section to articulate current limitations and outline directions for future research.
Response4:We sincerely thank the reviewer for their constructive suggestion. Accordingly, we have added a dedicated" Discussion"section before Section 4 (Conclusions) in the revised manuscript (see Lines 553–572) to provide a more comprehensive examination of the limitations of this study and outline directions for future work. This section primarily addresses the following three aspects: (1) the proposed method still has room for improvement in handling color fidelity under extreme lighting scenarios; (2) the class imbalance present in the dataset may affect the detection accuracy for minority fault categories (e.g., flashover); and (3) future work will aim to extend the core ideas of this framework to the defect detection of other power equipment to verify its generalizability. We believe this addition enhances the depth and forward-looking perspective of the paper. The discussion section added to the original article appears below.
Discussion
Although the proposed unsupervised image enhancement framework has demonstrated considerable effectiveness for multiple fault detection of insulators under complex lighting conditions, this study has certain limitations that point toward valuable future research directions.
Firstly, in a minority of extreme over-exposed or low-light scenarios, the enhanced images may occasionally exhibit slight color casts. This indicates that achieving better preservation of natural color fidelity while balancing luminance remains an area for further investigation.
Secondly, the dataset constructed for this study suffers from class imbalance, with "normal" insulator samples significantly outnumbering specific fault categories such as "flashover." While the image enhancement preprocessing improved the overall performance of the detection model, this skewed data distribution might still limit the model's ability to capture subtle features of the minority fault classes.
Finally, the current work focuses specifically on insulators. The core idea underlying our approach—improving the performance of subsequent detection tasks via unsupervised image enhancement—holds the potential to be transferred to the defect detection of other power equipment (e.g., surge arresters, instrument transformers). Verifying and extending the generalizability of this framework constitutes an important direction for future work.
Reviewer 2 Report
Comments and Suggestions for Authors
This reseach offers a mostly novel combination of techniques ( including grayscale-guided enhancement with luminance consistency loss) for insulators' fault detection under challenging lighting, which appears to be a new contribution in this domain. Improving fault detection under low-light / high-glare conditions is an important and practical problem. The content has significance for power line inspection and could be valuable to both researchers and practitioners.
The introduction is thorough and well-referenced, giving relevant context on insulator faults and existing detection methods, and clearly identifying the gap of uneven illumination.
The overall design (unsupervised enhancement module integrated with detection, plus synthetic data experiments) is well thought out to address the stated problem effectively.
The paper details the methodology (Section 2) with clear descriptions of the generator, grayscale attention mechanism, discriminator, and loss functions.
The experimental results (Section 3) are presented with appropriate tables, figures, and explanations. Ablation studies and comparisons to other methods are easy to follow and interpret.
Most figures and tables are informative with reasonable quality and resolution however some annotations are too small and hard to read.
There is a table heading typo “Figure 2.1” which doesn't exists.
The study would be stronger with more real-world validation. Additional tests on actual inspection images (beyond the synthetic extreme-lighting dataset) and analysis under various real lighting conditions would increase confidence in the method’s generalization to practical use.
Section 2 start with an unrelated template heading (“Mathematical model for valve cooling water temperature...???? ”) that should be removed or corrected. Also, there are small errors like a “Figure 2.1” reference. there are a few more minor errors. please check the entire manuscript for such issues.
It would help to discuss the computational efficiency of the enhancement step (e.g. can the U-Net + multi-patch discriminator run near real-time on UAV inspection data, or is there a speed vs accuracy trade-off?). Additionally, the authors should clarify why certain detection models (like YOLOv5m’s defect AP) showed little to no improvement or a slight drop with enhancement, analyzing these cases.
Comments on the Quality of English Language
The manuscript is written in generally clear and professional English. Aside from a few very minor typos, the language quality does not impede understanding.
Author Response
Reviewer 2
Comment 1.This reseach offers a mostly novel combination of techniques ( including grayscale-guided enhancement with luminance consistency loss) for insulators' fault detection under challenging lighting, which appears to be a new contribution in this domain. Improving fault detection under low-light / high-glare conditions is an important and practical problem. The content has significance for power line inspection and could be valuable to both researchers and practitioners.
The introduction is thorough and well-referenced, giving relevant context on insulator faults and existing detection methods, and clearly identifying the gap of uneven illumination.
The overall design (unsupervised enhancement module integrated with detection, plus synthetic data experiments) is well thought out to address the stated problem effectively.
The paper details the methodology (Section 2) with clear descriptions of the generator, grayscale attention mechanism, discriminator, and loss functions.
The experimental results (Section 3) are presented with appropriate tables, figures, and explanations. Ablation studies and comparisons to other methods are easy to follow and interpret.
Response to Comment 1:We sincerely thank the reviewer for their positive assessment of our work, recognizing it as a novel contribution in the domain of insulator fault detection under challenging lighting conditions with significant practical importance. Their appreciation for the introduction, overall design, methodology description, and presentation of experimental results is highly encouraging.
Comment 2.Most figures and tables are informative with reasonable quality and resolution however some annotations are too small and hard to read.
There is a table heading typo “Figure 2.1” which doesn't exists.
Response to Comment 2:We thank the reviewer for pointing out the issues with figures and formatting. We have taken the following corrective actions: (1) We have thoroughly checked all figures and ensured that all annotations and text are of sufficient size and clarity for easy reading. (2) We have corrected the erroneous reference to non-existent "Figure 2.1" and replaced it with the correct "Figure 1". The modified position is at line 109 of the original text. The modified original text is as follows
The overall structure is shown in Figure 1.
Comment 3.The study would be stronger with more real-world validation. Additional tests on actual inspection images (beyond the synthetic extreme-lighting dataset) and analysis under various real lighting conditions would increase confidence in the method’s generalization to practical use.
Response to Comment 3:We fully agree with the reviewer that additional real-world validation would significantly strengthen the study. In the revised manuscript, we have expanded Section 3.4.3 "Experimental test of image enhancement on actual power transmission lines" with more test results and analysis on actual inspection images (please see Pages 15-17, Tables 4 & 5, and corresponding discussions in the revised manuscript). These additions demonstrate the model's effectiveness under various real lighting conditions, further confirming its generalizability for practical use.
Comment 4.Section 2 start with an unrelated template heading (“Mathematical model for valve cooling water temperature...???? ”) that should be removed or corrected. Also, there are small errors like a “Figure 2.1” reference. there are a few more minor errors. please check the entire manuscript for such issues.
Response to Comment 4:We thank to the reviewer for the figure title issue. We have taken the following corrective actions:(1) We have removed the unrelated template heading "2. Mathematical model for valve cooling water temperature predictions" at the beginning of Section 2. And the title is revised to "2. Model overall structure design", the specific location is in Section 2, line 105 of the original text. (2) We have performed a careful proofreading of the entire manuscript to correct such formatting and referencing errors. The revised title looks like this.
- Model overall structure design
Comment 5.It would help to discuss the computational efficiency of the enhancement step (e.g. can the U-Net + multi-patch discriminator run near real-time on UAV inspection data, or is there a speed vs accuracy trade-off?). Additionally, the authors should clarify why certain detection models (like YOLOv5m’s defect AP) showed little to no improvement or a slight drop with enhancement, analyzing these cases.
Response to Comment 5:Although the overall mAP shows an improvement after introducing the illumination enhancement module (IEM), a slight decline in the AP of certain categories (e.g., the flashover class in YOLOv5) can still be observed. This phenomenon may result from the illumination enhancement altering the texture or brightness distribution of specific samples. In particular, for categories like flashover, whose visual features are mainly represented by high-brightness or overexposed regions, excessive enhancement may reduce the feature contrast between target and background, leading to a minor degradation in detection accuracy. Moreover, the IEM module focuses on global illumination correction rather than fine-grained texture recovery, which can slightly affect models that rely heavily on local feature cues.
Comment 6.Comments on the Quality of English Language
The manuscript is written in generally clear and professional English. Aside from a few very minor typos, the language quality does not impede understanding.
Response to Comment 6:We thank the reviewer for their overall positive comment on the language quality of the manuscript. We have performed an additional round of language proofreading to correct the remaining minor typos, ensuring the accuracy and fluency of the language. The revised part after proofreading is as follows:
Power transmission lines are a cornerstone of modern infrastructure, responsible for delivering electrical energy from generation sources to end users. As critical com-ponents, insulators electrically isolate conductors from supporting structures, thereby preventing leakage currents and maintaining the safety and stability of the grid. However, faults in insulators can significantly threaten the reliability of power systems. Such defects often arise from environmental contamination, mechanical damage, or flashover events. Therefore, systematic inspection and maintenance of insulators are essential to sustain their performance and ensure reliable operation under diverse en-vironmental conditions.
Deep learning, particularly in the field of computer vision, has significantly ad-vanced various industrial applications, including those in the power sector.
When processing large-scale datasets, researchers primarily focus on factors that influence detection accuracy, particularly complex backgrounds and multi-scale defects. Attention mechanisms are commonly employed to distinguish targets from background clutter, emphasizing salient regions while suppressing irrelevant information [7],[8]. For example, Zhou [9] integrated a Convolutional Block Attention Module (CBAM[10]) into YOLOv5, using max-pooling and average-pooling at both channel and spatial levels, and achieved a 98% average detection accuracy for glass insulator targets. Similarly, Hao [11] incorporated a CSPResNeSt backbone and a three-dimensional attention module (SiMAM[12]) into YOLOv4, improving insulator detection accuracy by 3.5% in complex back-grounds. However, these methods primarily focus on detecting intact insulators rather than identifying faults. Detecting small-scale fault regions remains challenging due to variations in target size resulting from differences in capture distance.
To address these challenges, this paper investigates the aforementioned issues in depth. First, we propose an insulator fault detection method for complex environments that integrates low-light enhancement and overexposure correction. The method mit-igates problems of uneven illumination by designing a deep-learning–based adversari-al network to preprocess raw images, thereby enhancing the visual quality of target regions. Furthermore, the object detection algorithm is optimized to handle complex backgrounds and small-scale targets, ultimately improving the accuracy of insulator fault detection under harsh environmental conditions. The main contributions of this work are summarized as follows:
It effectively mitigates the negative impact of complex lighting conditions on the qual-ity of transmission line inspection images, significantly enhancing the recognizability of target regions under both overexposed and low-light scenarios. Moreover, it pro-vides stable, high-quality image inputs that facilitate subsequent fault detection tasks. The following sections describe the detailed design and optimization process of the proposed model.
In the processing of transmission line inspection images, complex lighting conditions are primarily characterized by brightness imbalance caused by the coexistence of overexposed and low-illumination regions. This imbalance leads to the loss of local details in images, which in turn severely affects feature extraction and the accurate localization of fault regions.
Round 2
Reviewer 2 Report
Comments and Suggestions for Authors
Responses and amendments are satisfactory.
Comments on the Quality of English Language
The manuscript is written in generally clear and professional English.